# Single-cell and spatial transcriptomics enables probabilistic inference of cell type topography

Alma Andersson[1✉], Joseph Bergenstråhle[1], Michaela Asp ⓘ [1], Ludvig Bergenstråhle ⓘ [1], Aleksandra Jurek[1], José Fernández Navarro ⓘ [1] & Joakim Lundeberg ⓘ [1✉]

The field of spatial transcriptomics is rapidly expanding, and with it the repertoire of available technologies. However, several of the transcriptome-wide spatial assays do not operate on a single cell level, but rather produce data comprised of contributions from a – potentially heterogeneous – mixture of cells. Still, these techniques are attractive to use when examining complex tissue specimens with diverse cell populations, where complete expression profiles are required to properly capture their richness. Motivated by an interest to put gene expression into context and delineate the spatial arrangement of cell types within a tissue, we here present a model-based probabilistic method that uses single cell data to deconvolve the cell mixtures in spatial data. To illustrate the capacity of our method, we use data from different experimental platforms and spatially map cell types from the mouse brain and developmental heart, which arrange as expected.

[1] Science for Life Laboratory, Department of Gene Technology, KTH Royal Institute of Technology, Stockholm, Sweden. ✉email: alma.andersson@scilifelab.se; joakim.lundeberg@scilifelab.se

Techniques for spatial transcriptomics have advanced to a state where the entire transcriptome now can be spatially resolved; however, methods providing an exhaustive portrait of the expression with deep coverage do not yet guarantee resolution at the single-cell level[1–3]. Thus, transcripts captured at a given position may stem from a heterogeneous set of cells, not all necessarily of the same type. Hence, the observed expression profile at any location can be considered a mixture of transcripts originating from multiple distinct sources. Implicitly the presence of such composite profiles means that even though the transcriptional landscape can be thoroughly charted, the biological identity and spatial distribution of the cells generating this remains largely unknown.

As mentioned, spatial transcriptomics techniques face a dilemma of knowing the location of transcripts but not which cell that produced them. Conversely, single-cell RNA-sequencing experiments associate each transcript to an individual cell, but information regarding the positions of these transcripts within the tissue is lost. Given this set of complementary strengths and weaknesses, the notion of combining data from the two techniques to delineate the spatial topography of cell type populations is compelling.

Methods to deconvolve (bulk) RNA-seq data, informed by single-cell data, have existed for some time and could theoretically be applied to spatial data[4–6]. More recently, similar methods designed specifically for cell type deconvolution in spatial data have emerged and offered new biological insights. For example, the molecular characteristics of pancreatic ductal adenocarcinoma was thoroughly explored by such integration, testifying to the value of this approach[7]. However, these methods tend to exhibit certain limitations such as: only select cell types can be assessed, manual curation of data is required to form representative cell type "signatures", dependence on marker genes, or the results—usually some form of normalized score—lack a clear biological interpretation.

Here we first present a new alternative model-based method to integrate single-cell RNA-seq and spatial transcriptomics data, which utilize complete expression profiles rather than a select set of marker genes. Next, we use this method to spatially map cell types present in single-cell data originating from mouse brain and developmental heart onto corresponding tissue sections. Finally, we show how our approach outperforms others (designed for bulk RNA-seq deconvolution) when presented with synthetic data.

## Results

**Model description**. The framework we propose uses single-cell data to infer *proportion estimates* of each cell type at every capture location within the spatial data, eliminating any need for interpretation or annotation of abstract entities like factors or clusters upon analysis of the spatial data[8]. We consider the types' underlying expression profiles as inherent biological properties unaffected by the experimental method used to study them; meaning that certain information can be transferred between different data modalities, hence our use of single-cell data to guide the deconvolution process of the spatial data.

Our method rests on the primary assumption that both spatial and single-cell data follow a negative binomial distribution, commonly used to model gene expression count data, for a more rigorous discussion regarding the validity of this assumption see Supplementary Section 1.1 (ref. [9]). In single-cell data, observed expression values of a specific gene are taken as realizations of a negative binomial distribution where the first parameter (the rate) is a product between a scaling factor (to adjust for a cell's library size) and a cell-type-specific rate parameter common to all cells of the same type, and the second parameter (the success probability)

is only conditioned on gene and shared across all types. In the spatial context, gene expression values associated with a cell at any capture location is modeled similarly to the observations in single-cell data: the rates consisting of the same cell-type-specific parameters, but now adjusted for spot library size and bias between the experimental techniques; the gene-specific success probabilities are shared with the single-cell data without any modifications. Varying bias in experimental techniques is accounted for at a gene level, and treated as independent of cell type.

Since observations from the spatial assays we focus on represent sums of transcripts originating from multiple cells, not individual ones, this prompts for further expansion of the model. By virtue of the additive property among negative binomial distributions with a shared second parameter, the mixture of contributions—at a given capture location for a certain gene—also follows a negative binomial distribution of known character: the rate is equal to the sum of all the contributing cells' rates, while the success probability remains unaltered.

If the cell type and gene-specific parameters are known, deconvolving the spatial data is equivalent to finding the cell type population that most likely generated the observed gene expression values within each spatial location, for example by maximum likelihood or maximum a posteriori (MAP) estimation. Fortunately, these parameters can be estimated from single-cell data, where no mixing occurs, to then be used accordingly. We account for asymmetric data sets (when the cell type population in the single cell and spatial data do not match), by introducing an additional cell type in the deconvolution process, with flexible parameters that can adjust to the data. To briefly summarize our method, we first characterize each cell type's expression profile using single-cell data, then—within each capture location—find the combination of these types that best explains the spatial data, Fig. 1 outlines this procedure. For a more detailed description of the model, see "Methods".

By design, our method is compatible with any kind of spatial data where the observed transcription profiles consist of contributions from one or multiple individual cells. We will however mainly focus on the method's application to data originating from the technique presented by Ståhl et al. (referred to as ST), launched as the Visium platform and being one of the more accessible approaches to high-throughput spatial transcriptomics[3,10].

**Implementation**. We provide an (open-source) implementation of our model, *stereoscope*, which performs the deconvolution process and spatially maps cell types, see "Code Availability" for more details. Only three items are required to conduct the analysis, raw count matrices for (i) the single cell and (ii) spatial data together with (iii) annotations of the former. Due to the nature of our model-based method, normalization and other transformation procedures are not necessary, neither is gene selection (e.g., of highly variable or informative genes) a requirement. Any combination of single cell and spatial data sets of similar composition can be used, without the need for them to be paired (i.e., from the same tissue specimen), this allows publicly available resources to be utilized.

Two distinct steps constitute the implementation; first parameters of the negative binomial distribution are estimated from the single-cell data for all genes within each cell type. Equivalent parameters for a distribution describing the expression from a mixture of these types can be formed by a weighted combination of the single-cell parameters. In the second step said weights are estimated such that the resulting distribution provides the best

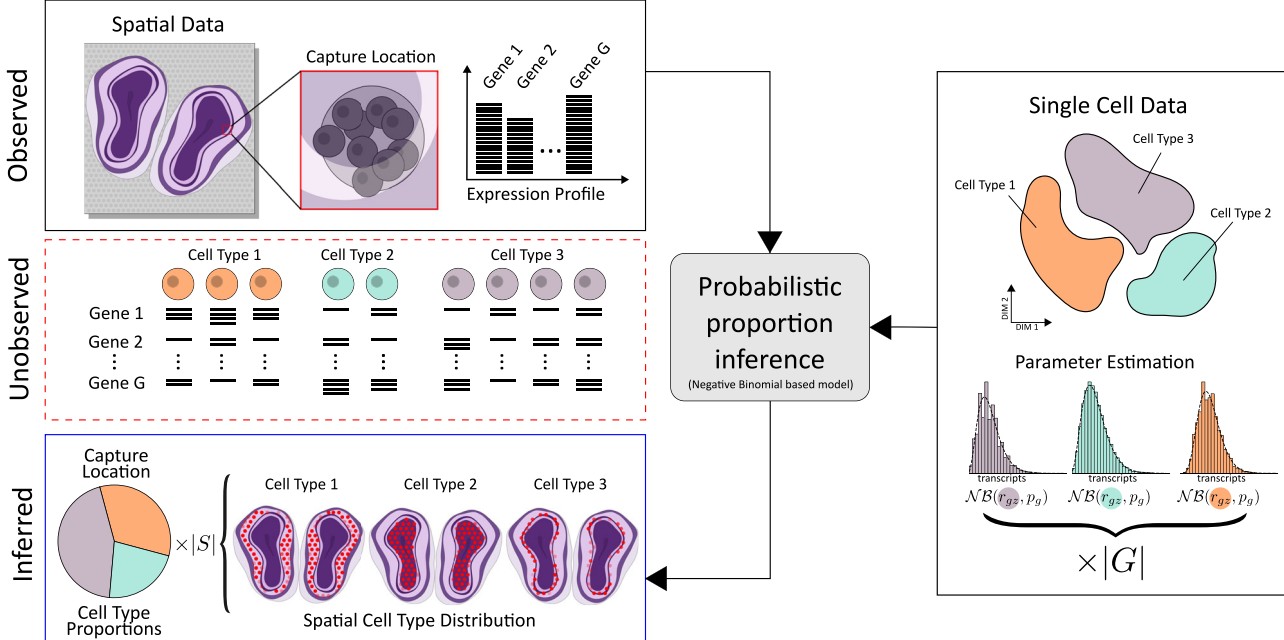

**Fig. 1 The observed expression profile at each capture location is a mixture of transcripts produced by one or multiple cells, where both the number and their types are unknown.** To model the unobserved cell population at a capture location, type-specific parameters are estimated from annotated single-cell data and combined to best explain the observed data for all |G| genes. This probabilistic model, based on the negative binomial distribution, enables inference of cell type proportions at each capture location; a procedure completely free from dependence on marker genes or gene set enrichment. Doing this for all |S| capture locations, results in a map over the spatial cell type landscape of the whole tissue.

explanation of the spatial data. Cell type proportions are obtained by normalizing the weights to make them sum to unity, see "Methods". Partitioning the process into two separate steps has the advantage that once the single-cell parameters have been estimated, they can be applied to any spatial data set of choice without the need to be re-estimated.

**Method application and evaluation**. In order to show the utility of our method we apply it to two different tissues: mouse brain and human developmental heart (6.5 post conceptional weeks, PCW). Furthermore, we only use spatial and single-cell sets derived from disparate sources to illustrate how paired data is not required to render factual results. See "Methods" for complete specifications of the data used. We consider the mouse brain and developmental heart tissues as good candidates to evaluate the method. The developmental heart's anatomy has been thoroughly explored and previous studies provide insights into the expected location of certain cell types. As for the mouse brain, it has also been extensively studied, resulting in plenty of resources describing its anatomical and molecular properties, one of them being the Allen Brain Atlas (ABA)[11]. By combining information of known cell type marker genes with the available in situ hybridization (ISH) data in ABA, the expected spatial distribution of these types can be deduced and used as a reference to compare our results. Figure 2 displays a subset of the results obtained upon mapping the single-cell data onto the mouse brain ST/Visium data sets (complete analysis in Supplementary Figs. 1–3). Each location is represented by a circular marker where the opacity of the face color indicates how abundant a certain cell type is at the given location, i.e. the higher the opacity, the higher the estimated proportion of the studied cell type (see "Methods"). As shown in Fig. 2a, single-cell clusters can be mapped onto the tissue, informing us of what spatial patterns they exhibit and how these clusters physically relate to each other—the spatial context may

also aid in assigning more distinct and descriptive identities to the clusters.

**Mouse brain analysis**. When assessing our results for the mouse brain hippocampal tissue, *Rarres2* is taken as a marker gene for ependymal cells (cluster 47), *Prox1* for dentate granule neurons (cluster 59), and *Wfs1* for pyramidal neurons (cluster 27). The resource (*mousebrain.org*) from which we accessed the single-cell data only provides broad classes like "Neurons" in its annotation, but observing the clusters' spatial arrangement enables us to assign them to more granular subtypes of these classes[12–14]. It is evident how the estimated proportions agree with the signals observed in the ISH experiments, confirming the proposed locations of these cell types. There is a high degree of consistency of the mapping between the different sections that are analyzed, speaking in favor of the method's robustness. In addition to coinciding with marker gene expression, the suggested spatial organization is further supported by already established knowledge regarding these types. Ependymal cells line the ventricular system, forming an epithelial sheet known as the ependyma, thus observing strong signals for this cell type in the lateral ventricular region is affirmative[15]. Dentate granule neurons reside within the dentate gyrus, a feature that our mapping manages to reproduce[16]. Pyramidal neurons belong to the broad class of excitatory neurons and populate regions such as the amygdala, cerebral cortex, and parts of Ammon's horn in the hippocampus, again in line with our results[17]. The usefulness of our method might be argued in a scenario where the marker gene(s) of types are known, since — in theory — expression levels could simply be visualized and used to infer the types' presence. However, due to the common presence of high sparsity and variance in spatial data, this single-gene approach does not always manage to re-create the patterns observed in ABA (see Supplementary Figs. 4–6), attesting to how using the full expression profiles of cell types

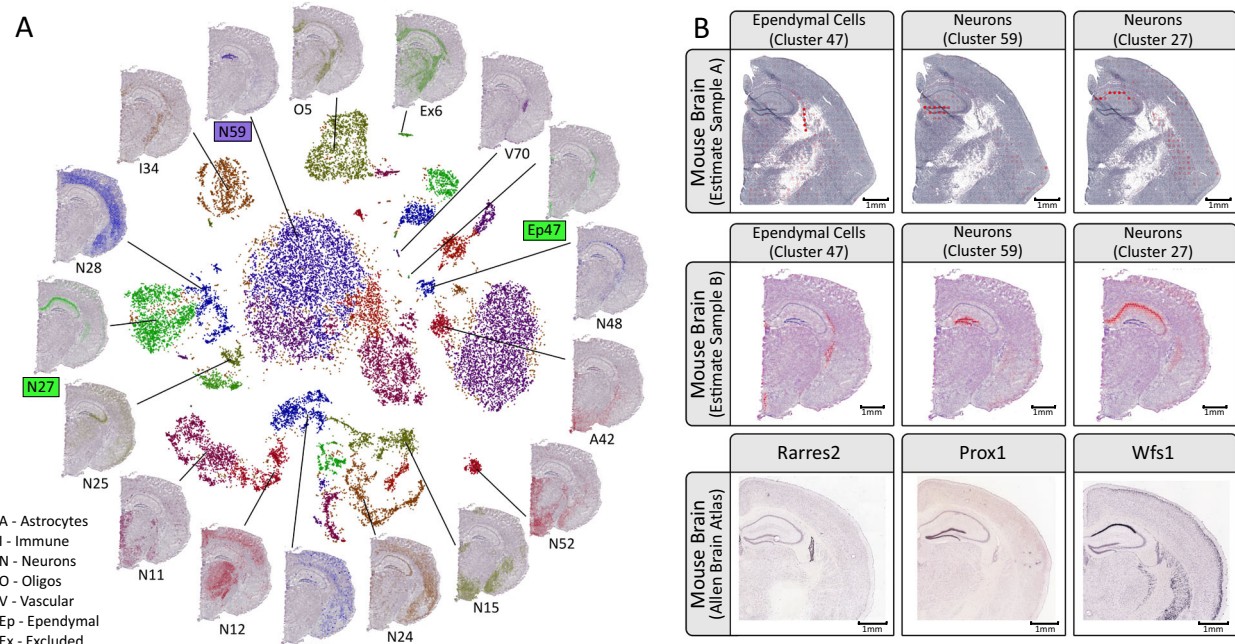

**Fig. 2 Mouse brain results overview. a** Visualization of the single-cell hippocampus data by using its gt-SNE embedding (inner region), with spatial proportion estimates of several clusters overlaid on the H&E-image (outer region) of sample mb-V1 (10× Visium array, 55 μm spots). The cluster labels are derived from the original single-cell data set (see "Methods")[24,31]. **b** Estimated proportions for 3 of the 56 clusters, here taken as cell types, defined in the mouse brain single-cell data set. Two different sections are used, mb-ST1 (ST array, 100 μm spots) and mb-V1, to illustrate the consistency between different array resolutions. Marker gene expression patterns obtained by ISH are found in the bottom row, taken from the Allen Brain Atlas. *Rarres2* is a marker gene of ependymal cells, *Prox1* for dentate granule neurons, and *Wfs1* for pyramidal neurons (the latter two both being subtypes of neurons). Face color opacity is proportional to the cell type proportion estimates; scale bars show 1 mm in respective image.

is preferable to relying on a few genes when working with these kinds of data.

**Developmental heart analysis**. In the developmental heart we observe how ventricular and atrial cardiomyocytes have the highest proportion values in the ventricular body and the atria, respectively, see Fig. 3. From the hematoxylin and eosin (H&E) images, blood cells are visible within the hollow cavities, the same areas as they are mainly estimated to reside within. Smooth muscle cells are almost exclusively mapped to the outflow tract, again, in concordance with their expected location[18]. Epicardial cells form a thin outer layer of the heart known as the epicardium, and this type is mainly assigned high proportion values in spots covering the edges of the heart[19]. Epicardium-derived cells arrange adjacent to the epicardial cells on the inner side of the heart in a somewhat thicker layer than the epicardium, and they are also known to be present in the outflow tract during its formation, a pattern recapitulated by our results[20].

**Additional experimental platforms**. To illustrate how the method may be used with other spatial techniques, we also analyze Slide-seq data from the hippocampus and cerebellum, where results from the technique's original publication are successfully reproduced. Cell types arrange similar to what was previously reported, and when aggregated re-creates the population landscape presented, see Supplementary Figs. 15–17. Using the same approach as the Slide-seq authors used to determine the number of "confidentially assigned" cell types to a capture location, we observe concordance with their results, see Supplementary Fig. 18.

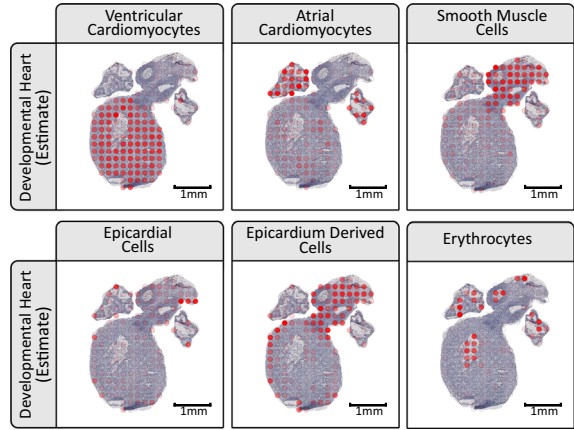

**Fig. 3 Excerpts of the estimated cell type proportions for the developmental heart, all from section dh-B.** The cell types presented are ventricular cardiomyocytes, atrial cardiomyocytes, smooth muscle cells, epicardial cells, epicardium-derived cells, and erythrocytes. For complete results see Supplementary Figs. 7–14.

**Cell type co-localization**. Once proportions have been estimated, subsequent analysis supplementary to visualization can be conducted. To exemplify, by looking at the spatial correlation between cell types (Pearson correlation on a per location basis) patterns of co-localization emerge, which may be informative and aid in elucidating which cell types that tend to interact or exhibit synergistic behavior. This is a complementary approach to those relying on receptor–ligand pairs to assess cell type interactions, without the need to curate lists of cognate receptors and ligands.

Results from this type of analysis when applied to the developmental heart can be seen in Supplementary Fig. 19.

**Comparison with alternative methods**. To enable comparison of our method with others, we devised a procedure to assemble synthetic data—resembling that obtained from the spatial techniques—from real single-cell data. Briefly explained, to generate synthetic data we randomly select cells from real single-cell data, and then add fractions of their expression values together. The exact proportions of cell types at each synthetic capture location can be computed since the identities of the sampled cells are known; hence, these datasets can be used as *ground truth* in comparative analyses. Two recently published methods (DWLS and deconvSeq) were used in the comparison; where our implementation outperformed both of these, see Supplementary Section 1.2 (refs. [6,21]).

## Discussion

In this study we formulate a probabilistic model to describe the relationship between single cell and certain spatial transcriptomics data, as a result we are able to develop a method that performs guided deconvolution of the mixed expression profiles present in the latter and thus spatially map the cell types identified in the former. When applied to real and synthetic data, our method presented results that agreed with previous literature and performed better than alternative approaches. Since the deconvolution process is guided by the single-cell data, the results (cell type proportions) are highly interpretable in contrast to those produced from completely unsupervised methods (e.g., factors or clusters), which tend to require further analysis to be annotated. Given how our method leverages information from the complete expression profiles, we see it as particularly attractive when working with complex tissues populated by several similar cell types, where mutually exclusive sets of marker genes are not guaranteed to exist.

Applications of our method are plenty: presence and identity of tumor-infiltrating immune cells could be assessed in cancers, or the types that constitute the tumor microenvironment charted; cell type interactions may be inferred from their spatial co-localization patterns; and enrichment of cell types within anatomical regions of interest determined, by examining how the proportion values are dispersed across the tissue. This list is far from comprehensive, but illustrates how information regarding the cell types' spatial distribution may serve as a basis for a multitude of different analyses. As a consequence of our two-step implementation, one may also conceive of a scenario where a database of fitted expression profiles (from the single-cell data) is constructed, making it possible to map cell types without having access to the raw count data as well as reducing the run time of the analysis.

Constant progress is made with respect to the experimental techniques, and while capture-based methods (e.g., Visium) currently do not guarantee single-cell resolution, one might envision this changing in the future. This would resolve the issue of mixed contributions to the observed gene expression, and eliminate the need to deconvolve data. Still, since the presence of a single cell can be considered a special case of a cell mixture (with one mixture component), we see a use of our method's ability to map types from one data modality to the other. For example, it could be used to ensure that the large efforts put into generation and annotation of single-cell atlases are not done in vain; given how our method would allow for type annotations to be transferred from single-cell data to the new and more highly resolved spatial data, guiding the process of characterizing the latter.

As outlined above, using single-cell data to guide the deconvolution has plenty of benefits, but also comes with certain limitations. One obvious example is how proportion estimates are only obtained for the cell types present in the single-cell data, and therefore statements regarding the spatial arrangement of cell populations are restricted to these types. Hence, it is preferable if the single-cell data to some extent is representative of the spatial data. This is especially true when large variance between tissue sections is expected or the cell type populations are highly specific to each individual (e.g., diseased tissues), in such cases paired data might be advantageous to use. As always, the character of the data largely dictates the quality of the results, for example; very shallowly sequenced single-cell data might not map as well as more deeply sequenced data if cell types are only distinguished by rare or lowly expressed genes. Still, issues related to discrepancies between the data sets are not unique to our method, but expected in any guided deconvolution approach. Fortunately, finding a suitable match for either data modality becomes increasingly easier as the number of public atlases (both spatial and single cell) continues to grow.

To conclude, we have presented a framework that enables spatial data to be deconvolved in a guided process and thus map cell types found in single-cell data onto a tissue. We have implemented this method in code, and release it as an open-source python package named *stereoscope* available at *github.com/almaan/stereoscope*. The procedure is seamless, transferable over multiple techniques, and does not require any pre-processing of the data.

## Methods

**Model**. The following notation will be used upon describing the model:

- $G$—the set of all genes
- $S$—the set of all spatial capture locations
- $Z$—the set of all cell types
- $C_s$—the set of all cells contributing to capture location $s$
- $n_{sz}$—number of cells from cell type $z$ at capture location $s$
- $x_{sg}$—counts of gene $g$ at capture location $s$
- $x_{sgc}$—counts of gene $g$ at capture location $s$ from cell $c$
- $z_c$—cell type of cell $c$
- $\alpha_s$—scaling factor at capture location $s$
- $\beta_g$—technique-based gene bias for gene $g$
- $r_{gz}$—rate parameter for cell type $z$ and gene $g$
- $p_g$—success probability parameter for gene $g$
- $|\cdot|$—cardinality of a given set
- $\boldsymbol{a}$—vector notation

Transcripts of a given gene ($g$) within a single cell ($c$) are taken as negative binomially distributed—with the rate ($r_{gz_c}$) being conditioned on a cell's type ($z_c$) and gene ($g$), while the success probability is only dependent on the gene in question (a common postulation)[9,22]. To account for certain technical biases, we also include a cell specific scaling factor $s_c$, set to the library size of respective cell. Thus we have

$$y_{gc} \sim \mathcal{NB}(s_c r_{gz_c}, p_g), \quad s_c = \sum_{g \in G} y_{gc}. \tag{1}$$

Values for the cell type specific parameters are then obtained by finding the MLE (maximum likelihood estimates), given the provided single-cell data. In the implementation this is achieved by taking the negative log-likelihood as an objective function to be minimized w.r.t. the parameters. We use PyTorch's autograd framework for the optimization, with *Adam* as an optimizer (default values are used for all parameters except learning rate, see below)[23].

In spatial data, the observable transcripts ($x_{sgc}$) of a given gene ($g$) from a cell ($c$) contributing to a specific capture location ($s$) are also considered negative binomially distributed, with the same conditioning as for the single-cell data. We assume that the efficiency by which certain genes are captured differs between the two techniques (spatial and single cell RNA-seq), what would be referred to as technique-based bias, and thus introduce a variable ($\beta_g$) to correct for this. A scaling factor ($\alpha_s$) for each spatial location is also included to account for technical variation between the spatial locations. The distribution used to model the spatial

data thus takes the form:

$$x_{sgc} \sim \mathcal{NB}(\alpha_s \beta_g r_{gz_c}, p_g). \qquad (2)$$

The total number of transcripts ($x_{sg}$) for a certain gene ($g$) at each spatial location ($s$) is simply the sum of observed transcripts from each cell ($c$) contributing to that spatial location, that is

$$x_{sg} = \sum_c x_{sgc}. \qquad (3)$$

With a shared second parameter ($p_g$) between all types ($z$), the first parameter exhibits an additive property and the total number of transcripts is negative binomially distributed as well:

$$x_{sg} \sim \mathcal{NB}\left(\sum_{c \in C_s} \alpha_s \beta_g r_{gz_c}, p_g\right). \qquad (4)$$

By introducing a quantity coefficient $n_{sz}$ representing the number of cells from a certain type ($z$) present at spatial location $s$, a change of index from *cells* to *types* is possible:

$$x_{sg} \sim \mathcal{NB}\left(\sum_{z \in Z} \alpha_s \beta_g n_{sz} r_{gz}, p_g\right). \qquad (5)$$

We then bundle the spatial location-specific parameters together in a *scaled quantity coefficient* ($v_{sz}$):

$$x_{sg} \sim \mathcal{NB}\left(\sum_{z \in Z} \beta_g v_{sz} r_{gz}, p_g\right), \quad v_{sz} = \alpha_s n_{sz}. \qquad (6)$$

Using vector notation this expression can be rewritten as

$$x_{sg} \sim \mathcal{NB}(\beta_g \mathbf{v}_s^T \mathbf{r}_g, p_g), \quad \mathbf{v}_s, \mathbf{r}_g \in \mathbb{R}^{|Z|}. \qquad (7)$$

To account for asymmetric data sets (where the cell types in spatial and single-cell data do not overlap perfectly) and noise we also include a form of "dummy" cell type, with gene specific rates ($\epsilon_g$) and a scaled quantity coefficient $\gamma_s$:

$$x_{sg} \sim \mathcal{NB}\left(\beta_g \mathbf{v}_s^T \mathbf{r}_g + \gamma_s \epsilon_g, p_g\right). \qquad (8)$$

If we define $w_{sz}$ as the normalized scaled quantity coefficients, excluding the noise capturing dummy cell type, that is:

$$w_{sz} = \frac{v_{sz}}{\sum_{z \in Z} v_{sz}} = \frac{\alpha_s n_{sz}}{\alpha_s \sum_{z \in Z} n_{sz}} = \frac{n_{sz}}{\sum_{z \in Z} n_{sz}}. \qquad (9)$$

This results in an expression which can be recognized as the proportion of each cell type at a given spatial location.

To avoid promiscuous assignment of explanatory power to the dummy cell type, we place a standard normal prior on all of its rates, i.e.

$$\epsilon_g \sim \mathcal{N}(0, 1). \qquad (10)$$

Cell type proportions ($w_{sz}$) are then taken as the MAP estimate of the distribution in Eq. (8) using the prior in Eq. (10), given the observed spatial data. Uniform priors are assigned to all other variables. More precisely, this is implemented by minimizing the negative logarithm of the posterior w.r.t. to the scaled quantities ($\{\mathbf{v}_s\}_{s \in S}$), the gene specific bias ($\boldsymbol{\beta}$), and parameters related to the dummy cell type ($\boldsymbol{\gamma}$ and $\boldsymbol{\epsilon}$, respectively). Similar to the procedure for single-cell data, the optimization is performed using PyTorch.

**Data processing**. Here we give a description of how the data are formatted and processed; note that our "starting material" are raw count matrices of single cell and spatial data. These matrices have cells or spatial locations along one dimension and genes along the other, with meta-data containing type annotations associated to the single-cell data. For exact details regarding how these count matrices were obtained from the raw sequencing data, we refer to their original publications.

*Gene selection*: Our method is not dependent on marker genes or curation of gene sets to be used during the inference; rather it is designed to use the complete expression profiles (all genes). Still, we noticed that using a subset (of reasonable size) of genes provide similar results to inclusion of all genes in the analysis, but with the benefit of reduced run-time. For all real data sets we therefore used the top 5000 highest expressed genes (total expression across all cells) in the single-cell data in our analyses.

More sophisticated criteria for selection of genes might enhance the performance, especially if effort is put into ensuring that marker genes for respective type are *included* in the subset. Given our claims of the method not necessitating gene list curation or knowledge of marker genes, we nevertheless deemed it appropriate to not incorporate such information in the process of selection, since this would be contradictory to our statement. Hence, the more simple "expression level"-based procedure.

*Human developmental heart*: The complete single cell data set provided in the paper "A spatiotemporal organ-wide gene expression and cell atlas of the developing human heart" was used to estimate the type parameters, hence resulting in a usage of 3717 cells distributed over 15 clusters[19]. Only the top 5000 highest

---

**Table 1 Synthetic data generation.**

Let $\mathcal{D}$ be an annotated single-cell data set;
Let $Z$ be the set of all types found in $\mathcal{D}$;
Let $\text{Idx}(z)$ be the indices of cells belonging to type $z \in \mathcal{D}$;
**For $s$ in** $1\ldots S$
  $C_s \sim \text{Unif}(lb, ub)$;
  $|Z_s| \sim \text{Unif}(1, |Z|)$;
  Let $Z_s$ be a subset of $Z$, consisting of $|Z_s|$ types formed by uniform sampling without replacement
  $p_{sz'} \sim \text{Dir}(1_s), \quad 1_s \in \mathbb{R}^{|Z_s|}, \quad z' \in Z_s$;
  $n_{sz'} = \lceil p_{sz'} \cdot C_s \rceil$;
  $w_{sz'} = n_{sz'} / \sum_k^S n_{kz'}$;
  Let $l_{sz'}$ be $n_{sz'}$ samples taken from $\text{Idx}(z')$ with equal probability and without replacement;
  $x_{sg} = \sum_{z' \in Z_s} \sum_{c \in l_{sz'}} \lceil \alpha \cdot y_{cg} \rceil$

---

expressed genes were used in the analysis. For the exact composition of the single-cell data set, see Supplementary Section 1.4.

The data were obtained from the same publication as the single cell data, using the eight sections from PCW 6.5. Only those spots under the tissue were used. From the 5000 genes selected in the single-cell data, the intersection of these and the complete set of genes found in the ST data was used.

*Mouse brain—hippocampus*: The single-cell data set was downloaded from *mousebrain.org*, where we used data with cells originating from Hippocampal tissue[24]. We first joined the "Class" and "Clusters" identifiers for each cell to form type labels. A subset of 8449 cells were sampled from the 29,519 cells found within the set. This subset was assembled by specifying both a global lower ($l$) and upper ($u$) bound for the number of cells to be included from each type, and then applying the procedure given in Eq (11) ($n_z$ representing the total number of cells from type $z$). We use an upper bound to reduce run time.

$$\begin{array}{lll} \text{Exclude cell type } z & n_z, \\ \text{Use all } n_z \text{ cells from } z & l \le n_z \le u, & (11) \\ \text{Sample } u \text{ cells from } z & u < n_z. \end{array}$$

The lower and upper bounds were set to 25 and 250 cells, respectively, giving the subset a total of 56 clusters. Only the top 5000 highest expressed genes were used in the analysis.

From the ST/Visium data, only those spots under the tissue were used. Three sections (mb-ST1, mb-ST2, and mb-V1) were used in the analysis. From the 5000 genes selected in the single-cell data, the intersection of these and the complete set of genes found in the ST/Visium data were used. mb-ST1 and mb-ST2 were analyzed together while mb-V1 was analyzed separately.

Slide-seq data were downloaded from Broad Institute's single-cell portal (*singlecell.broadinstitute.org*). We used the puck with ID 180413_7, taken from the project named "Slide-seq study". All beads with non-zero total counts were included in the analysis while from the 5000 genes selected in the single-cell data, the intersection of these, and the complete set of genes found in the Slide-seq data were used.

*Mouse brain—cerebellum*: As for the hippocampal tissue we downloaded single-cell data from *mousebrain.org* selecting "Cerebellum" as the tissue from which cells should originate. For this single-cell data we used the "Clusters" label. The same subsampling scheme as described above was used for this data, resulting in set of 7506 cells. Slide-seq data were downloaded from Broad Institute's single-cell portal (see above), more specifically the puck with ID 180819_11; all beads with non-zero total counts were included in the analysis.

See Supplementary Section 1.3 for more details regarding the composition of the sets.

*ISH images*: ISH images were downloaded from the Allen Brain Atlas. No modifications except for cropping were applied. References for the used images are:

- *Rarres2* (ref. [25])
- *Prox1* (ref. [26])
- *Wfs1* (ref. [27])

**Comparative analysis**. *Method*: To allow for performance comparison between methods we devised a procedure for generation of synthetic data. We decided to use a "semi-synthetic" approach not based on a negative binomial model, as this potentially could favor our model. Instead, single-cell data are used to assemble synthetic data with a structure similar to that of the spatial data our method is designed for. The procedure is described in Table 1.

Meaning that for every spatial capture location ($s$) we first sample the number of cells ($C_s$) contributing to this, and the number of types ($|Z_s|$) which these cells

**Table 2 Analysis parameters used for respective data set.**

|  | SC epochs | Spatial epochs | Learning rate | Top $N$ genes |
|---|---|---|---|---|
| Dev. heart | 50,000 | 50,000 | 0.01 | 5000 |
| Mouse brain[a] | 50,000 | 50,000 | 0.01 | 5000 |
| Synthetic data | 50,000 | 50,000 | 0.01 | 500 |

[a]The same single-cell parameter estimates (rates and logits) were used for all of the spatial data sets (ST, Visium, and Slide-seq), which were analyzed using identical settings.

may belong to. *lb* and *ub* represent the lower and upper bound, respectively, for the number of cells that are present at a spot. From the set of all types present in the data ($Z$), we then form a subset ($Z_s$) by uniform sampling of $|Z_s|$ types from $Z$ (without replacement). Unadjusted proportions ($p_{sz'}$) are then drawn from the probability simplex using a Dirichlet distribution (concentration set to 1 for all types). The actual number of cells from each type ($n_{sz'}$) is then set to the nearest integer number for the corresponding proportion of cells in the spatial capture location ($s$). The adjusted proportions ($w_{sz'}$) are given as the actual proportion based on the number of cells after nearest integer rounding. For every type ($z'$) present at a given spatial location, we then sample (without replacement) indices ($I_{sz'}$) of cells labeled as the same type within the single-cell data. To generate the expression value for a gene ($x_{sg}$) we sum the nearest integer approximation of the product between the single-cell expression values ($y_{sg}$) and a scaling factor ($\alpha$), a constant specified by the user, over all selected types and the sampled indices. By applying this procedure one obtains a data set with similar properties to that of the intended spatial data, but where the exact proportions are known.

*Generated sets*: Two synthetic data sets were generated by near identical procedures, the only difference being the values for upper ($ub$) and lower ($lb$) bounds, defined as in Table 1. For the first set, we let the range of cells present at a capture location be 10–30, which is representative of data originating from the ST technique. For the second set, we used a lower range of 1–10 cells, more in line with what is reported for the Visium platform.

The same single-cell data were used in the generation of both synthetic data sets, which is the hippocampus data taken from *mousebrain.org* (same as for the previous mouse brain analysis), we used the "Subclass" labels as annotations. We also subsampled the set according to the procedure described above (using 60 as lower and 500 as upper bound).

For each synthetic set, the subsampled set was split into two equally sized and mutually exclusive sets, i.e. sharing no cells. We refer to these as *generation* and *validation* sets. A synthetic spatial data set was then generated according to the procedure outlined in Algorithm 1 using the generation set as input. The resulting spatial data set consisted of 1000 spatial capture locations with expression values for 500 genes. The purpose of the validation set is to be used as the single-cell data used to deconvolve the spatial data by respective method.

*Comparison and evaluation*: To compare the performance between methods, we provided each of them with the validation single-cell data set and the generated synthetic spatial data to obtain proportion estimates for each spatial location. For each method we then computed the RMSE (Eq. 12) between the estimated proportions ($w$) and the ground truth ($\hat{w}$).

$$\text{RMSE}(\boldsymbol{w}_s, \hat{\boldsymbol{w}}_s) = \sqrt{\frac{1}{Z} \sum_z^Z (w_{sz} - \hat{w}_{sz})^2}. \quad (12)$$

Being interested in whether our method performed better than the others, we conducted an one-sided paired Wilcoxon signed-rank test to see whether the differences in RMSE values, in each capture location ($n = 1000$), were asymmetrically distributed around zero—in favor of our method. This was done using the R implementation of the Wilcoxon signed-rank test (*wilcox.test* : conf.int = TRUE, alternative = "less", paired = TRUE)[28].

Two published methods, designed for deconvolution of bulk RNA-seq data using single-cell data, were selected for comparison: *DWLS* and *deconvSeq*. The approach presented by Moncada et al. was not included due to very limited code availability and lack of a documented implementation.

*DWLS* treats the deconvolution task as an optimization problem cast in the form of a vector decomposition. The bulk RNA-seq data are represented by a vector ($t$) (of expression values) which is the product between a gene signature matrix ($S$) and a cell type number vector ($x$), see Eq. (13).

$$Sx = t. \quad (13)$$

The gene signature matrix ($S$) is static and the elements represent average gene expression values of marker genes from respective cell type derived from the single-cell data. The objective then becomes to find the optimal $x$, according to a weighted error function. The authors discard the more common OLS (ordinary least squares) approach in favor of a weighted scheme to account for rare cell types and ensure informative genes are not neglected during inference[6]. To conduct the analysis we downloaded the DWLS source code from https://github.com/dtsoucas/DWLS.

*deconvSeq* uses a negative binomial generalized linear model (GLM), with a log link for the mean. From the single-cell data (or pure bulk RNA-seq data), a projection matrix ($B_0$) is obtained by fitting a GLM to the data, the dispersion parameter is determined using edgeR and only conditioned on gene. The cell type proportions in a mixed sample are then estimated by finding the vector ($x$) that best fit the data when projected by $B_0$ onto the gene expression space of the top genes with the condition that all elements of $x$ are non-negative and sum to one[21]. To conduct the analysis we downloaded source code for deconvSeq from https://github.com/rosedu1/deconvSeq.

Slight modifications had to be made to the code in DWLS, though these changes did not concern the actual proportion estimation. All code used throughout the comparison, including wrappers for the methods when applying them to spatial data, are found in the github repository. The aforementioned modifications are accounted for in more detail at said repository.

To put the RMSE values into context, we compute the RMSE between probabilities drawn from a Dirichlet distribution (all concentration values set to 1) for an equal number of spatial locations as in the analyzed data sets. By repeating this for a select number of times ($n = 1000$), we obtain a "null-distribution" of RMSE values to compare the other RMSE distributions to it.

**Statistics and reproducibility**. Below, we describe specific details for the analysis of each pair of data sets, allowing the results to be properly reproduced (Table 2).

We used an one-sided Wilcoxon signed-rank test when comparing the performance between methods across all capture locations ($n = 1000$). We tested in favor of our model, i.e. if *stereoscope* on average has a lower RMSE. For more details see the "Methods" section "Comparison and evaluation".

In Supplementary Fig. 19 correlation values are called as significant ($p \leq 0.01$) or not; for exact computation of the $p$ values we refer to the documentation for the scipy (v.1.4.1) function *scipy.stats.pearsonr*, which was used (with default parameters) for this purpose[29]. The correlation values were computed across all eight developmental heart sections: in total, 1375 capture locations.

**Visualization and downstream analysis**. Scripts for all of the visualizations are provided at the github page.

*Proportions—separate visualization*: Upon visualizing the proportion of a single type within a given spatial location, the opacity of the face color corresponds to the estimated proportion. If nothing else is stated; proportion values are scaled within each section and cell type, to emphasize the spatial patterns. Such scaling is performed by dividing all proportion values of a certain cell type and section by the largest element within this set. No threshold or further adjustments of the values are applied after the scaling. When image data of the tissue are available, as for ST/Visium data, the array coordinates are transformed to pixel coordinates and the proportion estimates overlaid on the H&E image.

*Single-cell clusters (Fig. 2a)*: To generate the image presented in Fig. 2a, we used the coordinates obtained upon embedding the data within a two-dimensional manifold using gt-SNE[24]. These coordinates were provided in the single-cell data loom-file, as attributes named "_X" and "_Y", respectively, and hence were not generated by us. The cluster indices are those obtained upon joining the "Class" and "Clusters" identifiers for each cell. Clusters excluded from the proportion estimate analysis are not visualized in the gt-SNE plot. The proportion estimates are those obtained upon analyzing the mb-V1 section together with the single-cell data set as described in the section "Mouse brain—hippocampus".

**Cell type co-localization**. By computing the Pearson correlation (see Eq (14)) between each pair of cell types, treating each spatial location as a distinct data point, one obtains information regarding which cell types that share similar spatial distributions.

$$r(z_i, z_j) = \frac{\sum_{s \in S}(w_{sz_i} - \bar{w}_{z_i})(w_{sz_j} - \bar{w}_{z_j})}{\sqrt{\sum_{s \in S}(w_{sz_i} - \bar{w}_{z_i})^2}\sqrt{\sum_{s \in S}(w_{sz_j} - \bar{w}_{z_j})^2}}. \quad (14)$$

In Eq. (14) $z_i$ represents cell type $i$, the bar indicates the arithmetic mean, and $S$ is the set of spatial locations in the studied data set. Where $s$ represents a specific spatial location and $w_{sz}$ the proportion of cell type $z$ in said spatial location.

**Reporting summary**. Further information on research design is available in the Nature Research Reporting Summary linked to this article.

## Data availability

All data used in this manuscript are publicly available and can be found at original publications or repositories. We provide direct links to download all data in Supplementary Table 1.

## Code availability

The method is released as a tool named *stereoscope* available at https://github.com/almaan/stereoscope. Documentation for *stereoscope*, a tutorial, scripts used for visualization and further analyses are also found within the repository. In the tutorial we provide walk-through to reproduce some of the analyses presented in this paper. Since the code hosted at github might be updated as time progresses—to add new features or better align with user needs—we have deposited a "frozen" image of the current codebase, which will remain unaltered. This image can be found at https://doi.org/10.5281/zenodo.3951884 (ref. [30]).

The implementation and code for analysis are written in Python 3.7, the core functions rely on the following libraries (and built with versions): numpy 1.17.4, torch 1.3.1, scipy 1.4.1, and pandas 0.25.3. Additional libraries for tasks such as parsing and logging are used for the CLI application and visualization; the entire list is given at the github repository and included in the installation file.

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

## Acknowledgements

We want to thank Vilhelm Bohr, Tyler Demarest, and Deborah Croteau for providing us with early access to samples of Mouse Brain. Furthermore, we are grateful for the insightful comments and suggestions given by the reviewers, their contribution is immensely appreciated. Additional thanks are also given to The Knut and Alice Wallenberg (KAW) Foundation, the Thon Foundation, Erling-Persson Family Foundation, EU JPND INSTALZ, Foundation for Strategic Research (SSF), and Science for Life Laboratory and the Royal Institute of Technology (KTH) who enabled this work to be produced. Open access funding provided by Royal Institute of Technology.

## Author contributions

A.A. formulated the model, implemented it in code, and wrote the manuscript. J.F.N. and A.J. contributed with the mouse brain ST data. M.A. provided early access to the developmental heart single cell and ST data, aided in assessing the results obtained for the same tissue and commented on the manuscript. J.B., J.F.N., and L.B. gave comments on the methods and paper. J.L. supervised the project, commented on the paper, and provided computational resources. All authors have read and approved the text.

## Competing interests

The authors declare no competing interests.
