## [Peer Review File · Communications Biology]

Reviewers' comments:

Reviewer #1 (Remarks to the Author):

The manuscript presented by Andersson and colleagues presents a new computational tool for the deconvolution of spatial transcriptomics datasets using single-cell transcriptomics information. The work is well written and gives an understandable explanation of which problem the authors are trying to tackle, how they do it and the results obtained. The GitHub repository with the code and implementation is very neat and easy to follow, which can help a lot when people try to use this method.

Major comments:

1. The authors might consider starting the manuscript with the technical validation using synthetic spots. This seems to be the more logical order, before showing applications examples of brain and heart tissues.
2. Depending of the format limitations of the journal, I would prefer seeing the heart and benchmarking (synthetic spots) results as main figure.
3. The authors use non-paired data for their analysis and show good results on very well-known tissues with a very structured and established spatial architecture. They claim "Any combination of single cell and spatial data sets of similar composition can be used, without the need for them to be paired (i.e., from the same tissue specimen), allowing publicly available resources to be fully utilized." paragraph 5, sentence 2. I do not believe that claim will always necessarily hold, especially with diseased tissues.
4. The cluster annotation is poor for the mouse brain dataset, as shown in section 1.1.2 of the supplementary. I think labeling cells as Astrocytes_14 or Neurons_15 gives a good representation of what cell type it is. This is important in the spatial context since different cell sub-types are found to be in different spatial regions and knowing which cell type is mapped is important to assess if it is predicted to be in the correct spatial region. In Figure 2b, it is clear that the neurons they are mapping are CA1 and DG. The neurons mapped to the CA1 region are labelled as Pyramidal Neurons in paragraph 9, when there are other regions in the brain that also present this type of neurons (cerebral cortex and amygdala). In short, better annotation of the cell types would help in seeing if the predicted location is the right one.
5. It appears that the authors can differentiate between very similar cell types, but I would like to have the cell types annotated and then take a look at how good they are at differentiating between subtypes of a cell type, to see how specific one can go with this tool.
6. The authors do not mention at any point other methods to perform this spatial deconvolution for ST data - Itai Yanai publication. Ideally they would compare both methods. However, the reviewer is aware of the fact that the Yanai paper does not provide an implementation and has minimal code available.
7. The authors claim that gene expression data on the ST platform follow a negative binomial, but would it be possible to show this and verify the assumption?
8. The authors use the 5,000 most highly expressed genes, is there any benchmarking on why 5,000 most highly expressed, why not 3,000? Why not use the most highly variable genes, or cell type marker genes or a combination?
9. The authors compare their tool with the Slide-seq technology. Considering the fact that v2 is now published they could consider including the new data, which is of better quality. When reporting that they observe the same results on slide-seq data, could the authors show if they also get that ~65% of the beads correspond to a single cell and 35% to two cells. That to me would be a good indication that their method performs well on slide-seq data. The claim "To illustrate how the method may be used with other spatial techniques, we also analyze Slide-seq data from the hippocampal region, where results from its original publication are successfully reproduced (Supplementary Figures 16-18)" last paragraph before citations.

Reviewer #2 (Remarks to the Author):

The authors propose a probabilistic approach for the integration of spatial transcriptomics and single cell RNA-sequencing data. Interestingly the data are not required to be paired assuring a larger employment of the proposed method. The authors then present two examples of applications: human developmental heart and mouse brain. Finally the tool is provided as a Python package on github. The code is well documented and lot of examples of applications are provided.

The work is interesting, of high quality and relevant. The reviewer has only one minor request:

1. The method is based on the assumption that the expression of each gene can be correctly modeled as a negative binomial distribution, both in scRNAseq and in spatial transcriptomics. These authors are not the first making this assumption. However it would be interesting to test the quality of such approximation, especially on genes that are generally considered as markers of the various cell types. In addition, how much the results obtained by the tool are affected in case a different distribution is considered?

Reviewer #3 (Remarks to the Author):

Spatial transcriptomics assays can be understood as an extension of single cell transcriptomics assays where spatial information of the tissue organization is also captured. This allows for the study of tissue heterogeneity in both cellular as well as spatial context. Andersson et al.'s manuscript describes a probabilistic framework to study spatial data by leveraging cellular expression signatures from single cell datasets. Their approach falls under the general topic of deconvolution which has been used in the context of spatial transcriptomics earlier (see Soldatov et al., Moncada et al.) . However, the authors differ in their approach by presenting a fully probabilistic model for the underlying data and providing an easy to use interface for their software. This is a notable contribution to the literature but some points worth raising are provided below. Additionally, we were able to successfully download and run the software.

Major points:

- The Visium technology is fast evolving and as of this writing the per spot density is around 1-10 cells (depending on the tissue type). Can authors discuss, in this light, how useful will the deconvolution approach that they present be?
- Authors introduce the scaling factors α_s and β_g to accommodate for technical variation amongst spots and for aligning differing assays (ST versus SC). However, its not entirely clear if these parameters are identifiable given the formulation presented. Have authors considered demonstrating the utility of this via simulations, analytically or can justify it conceptually?
- Moreover, it isn't clear how useful the introduction of the scaling factor is in the deconvolution. As a gene-specific scaling parameter, β_g , is used – this implies around 5000 parameters for each gene (5000 is suggested value in the GitHub repo). It seems that this might cause the model to be too flexible and overfit. Have the authors compared model performance with and without scaling factors?
- The authors claim that no normalization is required for their package. However, to reiterate the earlier point, have the authors tried using just TPM values for both ST and SC expression data and not use scaling factors. Can they provide a justification for a more complex approach of using scaling factors and thus requiring no normalization?
- The authors do not provide any specifics on how the model is fit beyond saying that the

optimization was implemented in PyTorch. It would be a lot easier to assess the reliability, convergence and time complexity of the fitting algorithm if such details are made explicit.

- The authors present a simulation for spatial transcriptomics data; however, the simulation essentially produces independent spots and forgoes any spatial construction of the data. Moreover, the number of cells per spot are around 10 – 30. However, as mentioned before, Visium technology now has around 1 – 10 cells per spot. Can authors test for these slightly more complex scenarios?
- The manuscript is lacking in citing earlier literature which has used similar ideas of pairing single cell data with spatial transcriptomics assays. It would be informative if authors could place their approach in context of earlier published work (for example, Soldatov et al., Moncada et al. etc.)

Minor points:

- In equation (1), sc should be defined as the inverse of the quantity on the RHS.
- The choice of the representation of Negative Binomial seems non-standard. The NB distribution is either represented as $NB(n, p)$ or $NB(\mu, \phi)$. Can the authors explain the need of a hybrid representation in the form of $NB(\mu, p)$?
- Quotes in the document appear to be wrong – in LaTeX, please use ```` for the opening quotes.
- While it is appreciated that the authors release their software as a Github repo, it might be worth the effort to spruce up the document as some grammatical errors remain. Authors also state that the user requires a “cell x genes” matrix of single cell data to get started – however, the preprocessing step uses a loom format. Would it not be more streamlined if the authors provide an example starting with the “cell x genes” matrix itself?

References:

- [1] Soldatov R et al. Spatiotemporal structure of cell fate decisions in murine neural crest. *Science*, 364(6444), 2019.
- [2] Moncada R et al. Integrating microarray-based spatial transcriptomics and single-cell RNA-seq reveals tissue architecture in pancreatic ductal adenocarcinomas. *Nature Biotechnology*, 38(3):333–342, 2020.

Response to reviewers

2020-06-04

Reviewer	Number	Type	Comment	Response	Edit(s)	Line Numbers
1	1	Major	The authors might consider starting the manuscript with the technical validation using synthetic spots. This seems to be the more logical order, before showing applications examples of brain and heart tissues.	We thank the reviewer for the suggestion and are much inclined to agree that validation should precede application in the presentation of any workflow. We however consider the application of stereoscope to real data as a form of validation, where we assess the validity of the estimated proportions and implicitly the arrangement of the cell types by relating these to the expression of associated marker genes (Allen Brain Atlas) as well as previously published literature. To us, the synthetic data figures as a medium that facilitates comparison between methods, where the need for a "ground truth" is more explicit in order to quantitatively assess the performance of each method. We also wish to familiarize the reader with the method, before making a comparative analysis, hence why the comparison is introduced later in the paper - and as a consequence also the synthetic data. Based on this reasoning, we would like argue in favor of maintaining the current structure of the paper.	None	NA
1	2	Major	Depending of the format limitations of the journal, I would prefer seeing the heart and benchmarking (synthetic spots) results as main figure.	We are grateful for this suggestion and realize that inclusion of more results in the main text would enhance the reader's experience reading the manuscript. We have therefore added a figure (Figure 3) with excerpts from the developmental heart analysis, as to provide a complement to the information provided in the text and allowing the reader to better assess the validity of our claims. Given how the comparative analysis only plays a minor role in our study and the presented results are contained within a single sentence, we would like to argue that these results are better suited as supplementary figures rather than as a main figure.	Added Figure 3	Between line 115 and 116
1	3	Major	The authors use non-paired data for their analysis and show good results on very well-known tissues with a very structured and established spatial architecture. They claim "Any combination of single cell and spatial data sets of similar composition can be used, without the need for them to be paired (i.e. from the same tissue specimen), allowing publicly available resources to be fully utilized". paragraph 5, sentence 2. I do not believe that claim will always necessarily hold, especially with diseased tissues.	We appreciate that the reviewer highlight this and apologize for phrasing the statement in manner that might be interpreted as slightly arrogant. Hence, we have updated the passage that the reviewer referenced.	Updated Referenced Paragraph to: "Any combination of single cell and spatial data sets of similar composition can be used, without the need for them to be paired (i.e. from the same tissue specimen), this allows publicly available resources to be fully utilized. In some cases where high variance between tissue samples is expected or cell type populations are specific to each individual (e.g., diseased tissues), paired data may however be preferable."	46-50
1	4	Major	The cluster annotation is poor for the mouse brain dataset, as shown in section 1.1.2 of the supplementary. I think labeling cells as Astrocytes_14 or Neurons_15 gives a good representation of what cell type it is. This is important in the spatial context since different cell sub-types are found to be in different spatial regions and knowing which cell type is mapped is important to assess if it is predicted to be in the correct spatial region. In Figure 2b, it is clear that the neurons they are mapping are CA1 and DG. The neurons mapped to the CA1 region are labelled as Pyramidal Neurons in paragraph 9, when there are other regions in the brain that also present this type of neurons (cerebral cortex and amygdala). In short, better annotation of the cell types would help in seeing if the predicted location is the right one.	The reviewer is more than justified in pointing out the poor quality of the annotation associated with the mouse brain data set, we admittedly experienced quite some frustration with this during the compilation of the manuscript. Given how we did not produced this (single cell) data, and the only information available to us is that presented in the public resource (mouse.brain-map.org), we would like to argue that it is out of the scope of this paper to provide new (and better) annotations for the 56 clusters/types - especially since we cannot confirm that all of these can be related to previously identified and well characterized types. In addition, we are inclined to say that the clusters/types we use in this study do constitute what the reviewer refers to as "subtypes of cell types" - in this case subtypes of the more broad classes of Neurons, Astrocytes, Oligos etc.	None	NA
1	5	Major	It appears that the authors can differentiate between very similar cell types, but I would like to have the cell types annotated and then take a look at how good they are at differentiating between subtypes of a cell type, to see how specific one can go with this tool.	We would also like to address the comment regarding the cluster (Neurons 27, Pyramidal Neurons) that maps to CA1, which the reviewer (correctly) states is also supposed to map to other regions (amygdala and cerebral cortex). Our comment being that this cluster actually do exhibit higher proportion values, albeit not as high as in the CA1 region, in these regions (amygdala and cerebral cortex). Though we admit that our visual representation of the results might be suboptimal and fails to properly convey this at first glance, since one needs to "zoom" in on these regions to properly see these signals.		NA
1	6	Major	The authors do not mention at any point other methods to perform this spatial deconvolution for ST data - I believe Yanai publication. Ideally they would compare both methods. However, the reviewer is aware of the fact that the Yanai paper does not provide an implementation and has minimal code available.	We have considered to include discussion and reference to other methods, and fully agree that it's apt to reference to Yanai's publication. Hence, we have included a reference and more explicit mention of this publication within the main text. Due to the limited space, we are unable to provide a comparative analysis, in favor of other methods which have been released and presented as tools for deconvolution of gene expression data.	Updated text (LaTeX Format): (1) Main: "More recently, similar methods designed specifically for cell type deconvolution in spatial data have emerged and offered new biological insights. For example, the molecular characteristics of pancreatic ductal adenocarcinoma (PDAC) was thoroughly explored by such an integrative procedure, testifying to the value of this approach."(cite[14])" (2) Methods: We compared <code>textit{stereoscope}</code> with two methods designed for deconvolution of bulk RNA-seq data, both using single cell data to guide this process. The approach presented by Morcada et al. was not included due to very limited code availability and lack of a documented implementation.	(1) 33 - 42 (2) 506-509
1	7	Major	The authors claim that gene expression data on the ST platform follow a negative binomial, but would it be possible to show this and verify the assumption?	We appreciate the desire to validate this assumption given how multiple reviewers have posed near identical requests of further proof, and will therefore offer the same answer to all of these, being that we have put effort into providing what we believe is substantial support for making this assumption. First - since we only have a single observation of our NB-distributed variables (x_{ij}), making it hard to provide informed statements regarding the variables' distribution - we cluster the spots and assume that the expression of a gene follows the same distribution within all spots of a given cluster. Next we model the gene expression with three different distributions: Negative Binomial, Poisson and Normal. Finally we evaluate how well these distributions fit and manage to explain the data using the Bayesian Information Criterion (BIC) as well as visually inspecting the curves in relation to the empirical distributions, where the NB distribution outperforms the other two distributions. We present results for multiple clusters and several well known marker genes. This work resulted in the introduction of a new section to the supplementary data, which we refer to in the main text.	Updated text (LaTeX Format): (1) Main: "The method rests on the primary assumption that both spatial and single cell data follow a negative binomial distribution, commonly used to model gene expression counts. For more rigorous discussion regarding this assumption see Supplementary Section <code>ref{supp:nb}</code> , <code>cite{diseas2}</code> . Technical bias is taken as independent of cell type, and the types' underlying expression profiles are seen as inherent biological properties unaffected by the method used to study them." (2) Methods: Added Supplementary Section 1.3 "Characterization of Spatial Expression Data", this entails almost 20 pages and thus too large to be included in this table.	(1) 46 - 52 (2) 80-134 (page 26 to 46)
1	8	Major	The authors use the 5,000 most highly expressed genes, is there any benchmarking on why 5,000 most highly expressed, why not 3,000? Why not use the most highly variable genes, or cell type marker genes or a combination?	We realize that we failed to properly motivate method for selecting genes in the analysis, and can confirm that both of the reviewer suggestions by all means likely would enhance the performance of stereoscope. We have updated our text to better explain this, but we'd also like to provide a more elaborate answer to the reviewer. Since we claim that our method does not rely on knowledge of marker genes or careful selection of genes to be included - we decided to select our genes in the most simple way we could think of. By doing so we aimed to show that using a model-based approach where the expression profiles of cell types are used allows one to infer cell type proportions, despite certain marker genes potentially being excluded. In short, we didn't want to be accused of making false claims or not have the results to support our statements.	Added subsection to Methods (LaTeX Format): Our method is not dependent on marker genes or curation of gene sets to be used during the inference; rather it is designed to use the complete expression profiles (all genes). Still, we've noticed that using a subset (of reasonable size) of genes provide similar results to inclusion of all genes in the analysis, but with the benefit of reduced run-time. For all real datasets we therefore used the top 55000s highest expressed genes in the single cell data throughout the analysis. More sophisticated criteria for selection of genes might enhance the performance, especially if effort is put into ensuring that marker genes of respective type are <code>textit{included}</code> in the subset. Given our claims of the method not necessitating gene list curation or knowledge of marker genes, we nevertheless deemed it appropriate to not incorporate such information in the process of selection, since this would be contradictory to our statement. Hence, the more simple "expression level"-based procedure."	348 - 367
1	9	Major	The authors compare their tool with the Slide-seq technology. Considering the fact that v2 is now published they could consider including the new data, which is of better quality. When reporting that they observe the same results on slide-seq data, could the authors show if they also get that ~65% of the beads correspond to a single cell and 35% to two cells. That to me would be a good indication that their method performs well on slide-seq data. The claim "To illustrate how the method may be used with other spatial techniques, we also analyze Slide-seq data from the hippocampal region, where results from its original publication are successfully reproduced (Supplementary Figures 16-18)" last paragraph before citations.	We are very thankful for the suggestion of assessing the number of cell types that are assigned to the beads, in order to see whether our results match those presented in the Slide-seq publication. We have therefore processed and integrated additional single cell and Slide-seq data (Mouse Cerebellum). We used the cerebellum data since this was the type of tissue to which the reported numbers (approx 65 and 32%) refer to in the original publication. Once the results from stereoscope were obtained we implemented a procedure similar to that presented in the Slide-seq publication (using the ratio between the cell type proportion value and the L2 norm of the beads' proportion vectors) to call a cell type as confidently assigned to a bead. After doing this we obtained similar values 60.2 and 37.6%, respectively, where we believe the discrepancy between our values are due to us using a more granular single cell data set (more cell types). We hope that this new analysis strengthens the claim that our method is applicable to Slide-seq data. Since Slide-seq2 is still yet to be published in a peer-reviewed journal, we feel justified in not including this data in our study. Furthermore, Slide-seq data was included to show how our method could be applied to spatial data originating from other platforms than Visium or ST, and we would like believe that this point is equally well proven by using data from the first iteration with the Slide-seq technique as from the updated version - even though results might be better in the latter.	None	NA

Reviewer	Number	Type	Comment	Response	Editor(s)	Line Numbers
2	1	Major	The method is based on the assumption that the expression of each gene can be correctly modeled as a negative binomial distribution, both in scRNAseq and in spatial transcriptomics. These authors are not the first making this assumption. However it would be interesting to test the quality of such approximation, especially on genes that are generally considered as markers of the various cell types. In addition, how much the results obtained by the tool are affected in case a different distribution is considered?	We appreciate the desire to validate this assumption given how multiple reviewers have posed near identical requests of further proof, and will therefore offer the same answer to all of these, being that we have put effort into proving what we believe is substantial support making this assumption. First - since we only have a single observation of our NB-distributed variables (x_{ij}), making it hard to provide informed statements regarding the variables distribution - we cluster the spots and assume that the expression of a gene follows the same distribution within all spots of a given cluster. Next we model the gene expression with three different distributions: Negative Binomial, Poisson and Normal. Finally we evaluate how well these distributions fit and manage to explain the data using the Bayesian Information Criterion (BIC) as well as visually inspecting the curves in relation to the empirical distributions where the NB distribution outperforms the two other distributions. We present results for multiple clusters and several well known marker genes. Resulting in the addition of a new section to the supplementary data, which we refer to in the main text. Adding to the question of how different distributions would affect the result, we agree that from a theoretical standpoint, this is a truly interesting question; thus as mentioned above, we've compared how well different distributions describe the observed gene expression data. However, we have not developed and implemented additional models for the actual proportion inference, as we deem outside the scope of this paper.	Updated text: (1) Main: *The method rests on the primary assumption that both spatial and single cell data follow a negative binomial distribution, commonly used to model expression count data, for more rigorous discussion regarding this assumption see Supplementary 1.3 (2) Supplementary: Added Supplementary Section 1.3 "Characterization of Spatial Expression Data", this entails almost 20 pages and is thus too large to be included in this table.	(1) 46-52 (2) 80-134 (page 28 to 48)
3	1	Major	The Visium technology is fast evolving and as of this writing the per spot density is around 1-10 cells (depending on the tissue type). Can authors discuss, in this light, how useful will the deconvolution approach that they present be?	We thank the reviewer for this suggestion. It is very informative for the reader and makes our text more relevant; a paragraph has been added to the main text addressing this specific question.	Added to Main text: *Constant progress is made with respect to the experimental techniques, and while capture-based methods (e.g., Visium) do not yet guarantee single cell resolution, one might envision this changing in the future. An increased resolution would resolve the issue of mixed contributions to the observed gene expression, and eliminate the need to deconvolve data. Still, since presence of a single cell can be considered a special case of a cell mixture (with one mixture component), we see multiple venues of use for our method's ability to map types from one data modality to the other. For example, it could be used to ensure that the large efforts put into generation and annotation of single cell atlases are not done in vain; given how our method would allow for type annotations to be transferred from single cell data to the new more highly resolved spatial data, guiding the process of characterizing the latter.	129-137
3	2	Major	Authors introduce the scaling factors α s and β g to accommodate for technical variation amongst spots and for aligning differing assays (ST versus SC). However, it's not entirely clear if these parameters are identifiable given the formulation presented. Have authors considered demonstrating the utility of this via simulations, analytically or can justify it conceptually?	The reviewer raises some very valid and important questions regarding the introduction of scaling factors to account for differences between different assays, we have therefore attached a document and additional results/data in an attempt to provide a thorough and satisfying answer.	Evaluated and compared the analysis with and without scaling factors using synthetic data; see attached files for results. These results are not included in the manuscript. See below, or alternatively the file named rev-3-additional-response.zip	rev-3-additional-response.zip
3	3	Major	Moreover, it isn't clear how useful the introduction of the scaling factor is in the deconvolution. As a gene-specific scaling parameter, β g, is used - this implies around 5000 parameters for each gene (5000 is suggested value in the GitHub repo). It seems that this might cause the model to be too flexible and overfit. Have the authors compared model performance with and without scaling factors?			
3	4	Major	The authors claim that no normalization is required for their package. However, to reiterate the earlier point, have the authors tried using just TPM values for both ST and SC expression data and not use scaling factors. Can they provide a justification for a more complex approach of using scaling factors and thus requiring no normalization?	As the reviewer points out in the previous comments we consider both data types (SC and ST) as negative binomial (NB) distributed. Realizations from this particular distribution are positive integers, which in our context translates to raw count data. By applying TPM normalization, our observations would no longer pertain to the support of the NB distribution, and thus we wouldn't be able to use the distribution to model the data. The introduction of scaling factors occurs in relation to the parameter estimate, meaning that we are able to account for differences in library size (among cells) when estimating the rate-parameter. Furthermore, the scaling factors are fixed and based on a cell's library size, hence the number computations are less than that which a TPM normalization would require. We hope that these arguments provide sufficient justification for our approach.	None	NA
3	5	Major	The authors do not provide any specifics on how the model is fit beyond saying that the optimization was implemented in PyTorch. It would be a lot easier to assess the reliability, convergence and time complexity of the fitting algorithm if such details are made explicit.	We appreciate the suggestion and have updated our text accordingly, seeing how this increases transparency of the method.	Text updated to: *We use PyTorch's autograd framework for the optimization, with Adam as optimizer (default values are used for all parameters except learning rate, see below)	276-278
3	6	Major	The authors present a simulation for spatial transcriptomics data; however, the simulation essentially produces independent spots and forgoes any spatial construction of the data. Moreover, the number of cells per spot are around 1-30. However, as mentioned before, Visium technology now has around 1-10 cells per spot. Can authors test for these slightly more complex scenarios?	We have acted upon the reviewers wish to see a simulation where the cell numbers range between 1-10 cells, more representative of the Visium technology. These tests are included in a citation to Moncada et al. Supplementary. However, we have not included a spatial dependence between our data-points. We motivate this decision by the fact that neither stereoscope nor any of the methods we compare it with takes the spatial aspect into consideration, and therefore believe that not including this in our data would not favor/disfavor any of the methods.	(1) Text in methods updated to (LaTeX formatting included): *Two synthetic data sets were generated by near identical procedures, the only difference being the upper (β uS) and lower (β lS) bounds described in Algorithm 1 (ref[algo:synthetic]). For the first set, we let the range of cells present at a capture location be β 10-30S, which is representative of data originating from the original ST technique. For the second set, we used a lower range of β 1-10S cells, more in line with what is reported for the Visium platform. The same single cell data was used in the generation of both synthetic data sets; which is the hippocampus data taken from textit[mousebrain.org] (same as for the previous mouse brain analysis), we used the "Subclass" labels as annotations. We also subsampled the set according to the procedure described above (using β 60S as lower respectively β 500S as upper bound). For each synthetic set: The subsampled set was split into two equally sized and mutually exclusive sets, i.e. sharing no cells. We refer to these as 'test(1)generation' and 'test(1)validation' sets. A synthetic spatial data set was then generated according to the procedure outlined in Algorithm 1 (ref[algo:synthetic]) using the generation set as input. The resulting spatial data set contained β 1000S spatial locations and β 500S genes (highest total expression). The purpose of the validation set is to be used as the single cell data provided together with the spatial data as input to respective method.* (2) Added Figure and Table to Supplementary section 1.2.3 Comparison	(1) 477-504 (2) Between line 78 and 80
3	7	Major	The manuscript is lacking in citing earlier literature which has used similar ideas of pairing single cell data with spatial transcriptomics assays. It would be informative if authors could place their approach in context of earlier published work (for example, Soldatov et al., Moncada et al. etc.)	We have updated our manuscript to, hopefully, better put our contribution into the context of already existing methods within the realm of integrating single cell and spatial data; and doing so included a citation to Moncada et al. given its demonstrated use on ST data. We have not cited Soldatov, given how the publication in concerned with ISS data which are significantly different from ST/Visium and not discussed in the paper.	1. Main text updated to (LaTeX Format): *Methods to deconvolve (bulk) RNA-seq, informed by single cell data, have existed for some time and could theoretically be applied to spatial data (see stereoscope, cell2cell). More recently, similar methods designed specifically for cell type deconvolution in spatial data have emerged and offered new biological insights. For example, the molecular characteristics of pancreatic ductal adenocarcinoma (PDAC) was thoroughly explored by such an integrative procedure, testifying to the value of this approach (cite[la]). However, these methods all tend to exhibit certain limitations such as: only select cell types can be assessed, manual curation of data is required to form representative cell type "signatures", dependence on marker genes, or the results - usually some form of normalized score - lack a clear biological interpretation. Since our method is model-based and utilize complete expression profiles rather than select set of genes, we are able to avoid these issues. As a consequence, we consider our method especially attractive when working with complex tissues populated by multiple similar cell types, where mutually exclusive sets of marker genes are not guaranteed to exist.*	(1) 35-42 (2) 160-172
3	8	Minor	In equation (1), sc should be defined as the inverse of the quantity on the RHS.	We are very thankful for bringing this to our attention - as made clear by the comment, the text and equation do not agree. It is however the text that is incorrect and not the equation. We have updated the text accordingly.	Text Updated to (LaTeX Format): 10 account for certain technical biases, we also include a cell specific scaling factor β l_cS, taken as the library size of respective cell."	268-270
3	9	Minor	The choice of the representation of Negative Binomial seems non-standard. The NB distribution is either represented as $NB(r, p)$ or $NB(\mu, \phi)$. Can the authors explain the need of a hybrid representation in the form of $NB(\mu, \beta)$?	It is true that other tools/methods that use the Negative Binomial distribution to model expression data use the common "mean and dispersion" parameterization, which is preferable when estimates of means such as log-fold change are desired. Our representation is nevertheless not uncommon in other applications, and is even often given as the standard representation in literature. For an example see the widely acknowledged Statistical Distributions 4th Edition by Forbes et al (page 139, ISBN 978-047039634). For a slightly more accessible but also less formal example we refer to the wikipedia page of said distribution. The parameterization we use is also equivalent to the one provided by both the PyTorch framework as well as TensorFlow. We hope that this explanation is satisfying and sufficient to answer the reviewer's questions.	None	NA
3	10	Minor	Quotes in the document appear to wrap - in LaTeX, please use "" for the opening quotes.	We did not catch this mistake ourselves, somewhere in a conversion between formats, something must have gone wrong. It's very kind to point this out - the issue has been addressed and is now resolved. We thank the reviewer for bringing the linguistic flaws of the software documentation to our attention, we have done our best to correct any grammatical errors found in the documentation.	Updated all quotes to proper format.	NA
3	11	Minor	While it is appreciated that the authors release their software as a GitHub repo, it might be worth the effort to spruce up the document as some grammatical errors remain. Authors also state that the user requires a "cell x genes" matrix of single cell data to get started - however, the preprocessing step uses a loom format. Would it not be more streamlined if the authors provide an example starting with the "cell x genes" matrix, itself?	The reason for us using a loom file as starting material was to show how all steps of the analysis presented in this study were executed, as to have maximal transparency and make it easy to reproduce our results. We also included the output from processing this loom file in our repo, but realize that we can be more clear with conveying that such "regio" data exists and that certain pre-processing steps may be skipped in the vignette.	GitHub documentation updated	NA

Additional response to Reviewer 3 comments 2 and 3:

Exact Comments :

Authors introduce the scaling factors α_s and β_g to accommodate for technical variation amongst spots and for aligning differing assays (ST versus SC). However, its not entirely clear if these parameters are identifiable given the formulation presented. Have authors considered demonstrating the utility of this via simulations, analytically or can justify it conceptually?

Moreover, it isn't clear how useful the introduction of the scaling factor is in the deconvolution. As a gene-specific scaling parameter, β_g , is used – this implies around 5000 parameters for each gene (5000 is suggested value in the GitHub repo). It seems that this might cause the model to be too flexible and overfit. Have the authors compared model performance with and without scaling factors?

Conceptual Justification

The scaling factors (β_g) are introduced to our model to account for eventual differences in capture efficiency of genes between single cell and spatial platforms. The scaling factors are defined on a per gene basis, meaning they are agnostic to the notion of cell types and spatial location. While polyA capture is used in both assays, this occurs under vastly different circumstances. In single cell assays, cells and the vector to which the capture probes are linked tends to be physically separated from other such pairs (e.g., in the Chromium platform, GEM beads and cells are assorted into droplets), where cells are lysed in order for transcripts to hybridize with the polyT sequence of the capture probes. In spatial data, the tissue is not generated and transcripts need to migrate from cells down to the surface of the array (ST/Visium) or beads (Slide-seq), for the hybridization to occur. Hence, the environment within the spatial assay is physically crowded more and less “pure” than that of the single cell assay. In addition, the reagents used differ to some degree between the two platforms during the experimental procedure. Taking this into consideration, we believe it's fair to assume that the extent to which certain transcripts are captured, differ between the two assays. For example, some transcripts may be more prone to interact with the surrounding environment (e.g., proteins and cell debris), thus reducing their tendency to be captured by the probes when compared to single cell assays. Treatment with different regents and distinct experimental protocols might also activate/deactivate various biological processes (not necessarily confined to specific cell types), meaning the relative expression levels between transcripts would differ between the two assays. These are our main conceptual arguments for the inclusion of the scaling factors.

Computational Evaluation

The reviewer, rightfully, questions the value of introducing this scaling parameter into our model, and whether it has any effect on the performance of the deconvolution process. To answer this question, we used the synthetic data (1-10 cells per capture location), generated for the comparison between existing methods and stereoscope, to compare the two alternatives: (i) inclusion of gene scaling factors as parameters to be learnt during optimization and (ii) no scaling factors. In the stereoscope tool, we've enabled this option by inclusion of a flag “--beta_freeze”, which effectively freezes all scaling factors to 1. When evaluating the results from

respective model when applied to the synthetic data, it's obvious that the scaling factors enhances the performance, see the image below (same format as Supplementary Figure 22-23)

To support our reported results, we also attach the proportion estimates from each run with a stereoscope.

Runtime and Flexibility

We fully agree with the statement that more parameters to be estimated will have an impact on runtime, but given how it enhances performance (above) we consider this acceptable. Furthermore, stereoscope is not designed as a tool to be run iteratively on the same data set tweaking the different analysis parameters (which there are few of) to see how the result changes, but rather as a one time analysis where run-time is not of highest priority as long as it is within a reasonable time range. Of course a faster analysis is always preferable, but we consider it second to result accuracy.

Regarding the risk of eventual overfitting, we believe that this question is partially addressed by the preceding results, but to give a complete answer. To us, the results from the synthetic as well as real data provide substantial support that our model works and provides meaningful proportion estimates; hence, the scaling factors do not seem to introduce too much flexibility, making the model behave unpredictably.

Images

Below are the newly introduced images, none of the other images were updated. We refer to the main and supplementary text to see the images in their actual context.

Cluster : 0

Cluster : 1

Cluster : 2

Cluster : 3

Cluster : 4

Cluster : 5

Cluster : 6

Cluster : 7

Cluster : 8

Cluster : 9

Cluster : 10

Cluster : 11

Cluster : 12

Cluster : 13

Cluster : 14

- cluster.name
- cluster 0
 - cluster 1
 - cluster 10
 - cluster 11
 - cluster 12
 - cluster 13
 - cluster 14
 - cluster 2
 - cluster 3
 - cluster 4
 - cluster 5
 - cluster 6
 - cluster 7
 - cluster 8
 - cluster 9

Slc6a1

Grin1

Gja1

Snap25

Scn2b

Gabra1

Aldoc

Calb1

Apod

Eno2

Olig1

Tubb3

REVIEWERS' COMMENTS:

Reviewer #1 (Remarks to the Author):

Overall the authors responded successfully to most of the comments made by the reviewers.

Minor comments:

Validation using synthetic spots could use classification metrics along with the already provided RMSE. This would allow them to further validate the predictions.

Have the authors checked the annotations in the metadata file provided at the download website: <http://mousebrain.org/downloads.html>

Plots provided to illustrate NB being the best distribution to fit the data lack legends.

When claiming the method could be used for publicly available atlases, have they checked the model's performance on shallowly sequenced datasets?

Reviewer #2 (Remarks to the Author):

The authors addressed the concerns of this Reviewer. The paper is now good for publication.

Reviewer #3 (Remarks to the Author):

I am satisfied with the authors revisions, which are very comprehensive and address our concerns.

Reviewer	No.	Comment Type	Comment	Answer	Modifications
1	1	Major	Overall the authors responded successfully to most of the comments made by the reviewers.	We are happy to hear that we were able to answer the major concerns of this reviewer, and want to thank him/her for the time dedicated to assess our manuscript thoroughly and with a critical mind. The comments we received were very valuable and significantly improved the character of the manuscript, for this we are more than grateful.	N/A
1	2	Minor	Validation using synthetic spots could use classification metrics along with the already provided RMSE. This would allow them to further validate the predictions.	We are not completely sure of exactly which "classification metrics" that reviewer has in mind here, or the specific analysis that would allow for such metrics to be calculated. Our method is designed to deconvolve observations with mixed contributions from (potentially) multiple cell types, to generate a form of "soft labels" (probability estimates) - a task more challenging (e.g., due to overlapping marker genes or similar expression profiles) than had the capture locations only consisted of one unique cell type, which then could be assigned "hard labels" (one spot, one type). Of course synthetic data where the observations are composed of only one cell type could be generated, and used to evaluate the methods' accuracy, precision and recall etc. - we however would argue that this does not evaluate the performance of the different methods with respect to the task they were designed for, as it ignores any notion of relative abundance among the cell types within each capture location, something the RMSE is better designed for. Furthermore, observations that are 'pure' with respect to their cell type composition are not representative of the data we have designed our method for, nor the data which the other methods were designed for, hence we are not fully sure that this classification analysis would help the reader to gauge how useful the respective methods are. If the reviewer had a specific type of analysis in mind which we fail to consider in the argumentation above, we apologize and would be more than willing to conduct this, if those insights are shared with us.	None
1	3	Minor	Have the authors checked the annotations in the metadata file provided at the download website: http://mousebrain.org/downloads.html	We have indeed checked the referenced file at mousebrain.org, however we fail to see which material in this file that the reviewer wants us to pay extra attention to, alternatively feel as if we have overlooked. However, if it is a question of working with finer tiers of cell type annotations in order to validate our method; we would be inclined to say that by showing how cell types in the developmental heart map according to their expected positions, how certain cell types in the mouse brain map to what can be assumed to be their expected areas (consistently across platforms), as well as managing to reproduce previously published mappings in the mouse brain (Slide-seq results), the claims regarding our method's performance are fairly well motivated. Still, if there is something we have missed that the reviewer would like us delve further into, a pointer to this would be much appreciated and we will do our best to pertain to these wishes.	None
1	4	Minor	Plots provided to illustrate NB being the best distribution to fit the data lack legends.	We thank the reviewer for pointing this out, it was a mistake from our side which now has been corrected.	Legends have been added to Supplementary Figures 22-37.
1	5	Minor	When claiming the method could be used for publicly available atlases, have they checked the model's performance on shallowly sequenced datasets?	This is a very valid question, to which the answer is no; we have not systematically evaluated how the sequencing depth of the data might affect the result. We also realize that the way we expressed ourselves could be considered slightly arrogant or exaggerated, this was never our intention as we want to convey a truthful image of our method, thus we have included the following paragraph into the Discussion section of the main text, in hope to correct our mistake: "As always, the character of the data largely dictates the quality of the results, for example; very shallowly sequenced single cell data might not map as well as more deeply sequenced data if cell types are only distinguished by rare or lowly expressed genes. Still, issues related to discrepancies between the data sets are not unique to our method, but expected in any guided deconvolution approach. Fortunately, finding a suitable match for either data modality becomes increasingly easier as the number of public atlases (both spatial and single cell) continues to grow." We believe this addresses limitations with our method in the context of the objective it was designed for, hopefully the reviewer is of the same opinion and finds this correction satisfying.	Paragraph added to Main text Discussion: "As always, the character of the data largely dictates the quality of the results, for example; very shallowly sequenced single cell data might not map as well as more deeply sequenced data if cell types are only distinguished by rare or lowly expressed genes. Still, issues related to discrepancies between the data sets are not unique to our method, but expected in any guided deconvolution approach. Fortunately, finding a suitable match for either data modality becomes increasingly easier as the number of public atlases (both spatial and single cell) continues to grow."
2	1	Major	The authors addressed the concerns of this Reviewer. The paper is now good for publication.	We thank this reviewer for making us thoroughly assess the validity of our assumptions regarding the character of the spatial data; to us, this not only enhanced the rigour of the manuscript but also makes it more attractive for the reader given how the assumptions now are backed up by stronger evidence.	N/A
3	1	Major	I am satisfied with the authors revisions, which are very comprehensive and address our concerns.	We want to express our gratitude towards the reviewer who clearly spent a lot of time to make a deep and just assessment of our method and the theory behind it; many of the questions and comments that were posed had an immediate (positive) impact on the quality of our manuscript's content, and all of them forced us to properly think through our argumentation and claims. Many improvements occurred as a consequence of the first review round, and thereviewer played a major role in this.	N/A